# Current Treatment Methods for Charcot–Marie–Tooth Diseases

**DOI:** 10.3390/biom14091138

**Published:** 2024-09-09

**Authors:** Hongxian Dong, Boquan Qin, Hui Zhang, Lei Lei, Shizhou Wu

**Affiliations:** 1Department of Orthopedic Surgery and Orthopedic Research Institute, West China Hospital, Sichuan University, Chengdu 610041, China; 2022324020044@stu.scu.edu.cn (H.D.); 2021324025286@stu.edu.cn (B.Q.); zhanghui1437@wchscu.cn (H.Z.); 2State Key Laboratory of Oral Diseases & National Clinical Research Center for Oral Diseases, West China Hospital of Stomatology, Sichuan University, Chengdu 610041, China

**Keywords:** Charcot–Marie–Tooth, pathomechanism, rehabilitation, surgery, drug therapy, gene therapy

## Abstract

Charcot–Marie–Tooth (CMT) disease, the most common inherited neuromuscular disorder, exhibits a wide phenotypic range, genetic heterogeneity, and a variable disease course. The diverse molecular genetic mechanisms of CMT were discovered over the past three decades with the development of molecular biology and gene sequencing technologies. These methods have brought new options for CMT reclassification and led to an exciting era of treatment target discovery for this incurable disease. Currently, there are no approved disease management methods that can fully cure patients with CMT, and rehabilitation, orthotics, and surgery are the only available treatments to ameliorate symptoms. Considerable research attention has been given to disease-modifying therapies, including gene silencing, gene addition, and gene editing, but most treatments that reach clinical trials are drug treatments, while currently, only gene therapies for CMT2S have reached the clinical trial stage. In this review, we highlight the pathogenic mechanisms and therapeutic investigations of different subtypes of CMT, and promising therapeutic approaches are also discussed.

## 1. Introduction

Charcot–Marie–Tooth (CMT) disease, also known as hereditary motor and sensory neuropathy (HMSN), is a genetically heterogeneous group of hereditary peripheral nerve diseases, mainly affecting peripheral axons and Schwann cells, with an estimated prevalence of 1:2500 individuals [1]. Originally, patients with CMT were described by Virchow, Eulenburg, Friedreich, Osler, and others as those with a disorder of peroneal muscular atrophy until, in 1886, Jean-Martin Charcot and Pierre Marie and Henry Tooth independently described this neuropathy, and the name CMT has since been used [2]. As a length-dependent motor and sensory neuropathy, the disorder features a slowly progressive muscular atrophy, starting in the feet (presenting as pes cavus or hammertoes) and legs and moving to the upper extremities (Figure 1), with segmental vasomotor disturbances, while the proximal muscles and muscles on the trunk, shoulders, and face have relatively high integrity [3]. These are the usual phenotypes of the disease, as there are also some CMT types with much more severe phenotypes.

By detection of specific alleles, mutations, genotypes, or karyotypes, genetic testing can also be used as a method for disease determination. For example, CMT1A usually detected with MLPA (multiplex ligation-dependent probe amplification) is caused by mutations in the peripheral myelin protein 22 gene (*Pmp22*), most often duplications and less commonly point mutations, which may cause a severe phenotype [4]. With the clinical application of genetic testing in disease assessment, CMT diagnostic methods were simplified, and decision-making processes in neuropathy were improved [4,5]. High-throughput next-generation sequencing (NGS) technologies allow for thousands to billions of DNA fragments to be simultaneously and independently sequenced [6]. With NGS approaches, novel genes attributing to CMT development have been uncovered, and our knowledge of the pathogenic mechanisms behind CMT is improving. The further sub-classification of CMT based on genetic heterogeneity has been conducted with the NGS approach.

Although oral pain killers and supportive, rehabilitative, and surgical regimes may be helpful in the management of symptoms, the gene-specific treatments in current development are the most promising treatment modalities for neuropathy cures, for example, antisense oligonucleotide (ASO) treatments [7]. Here, we review the pathogenic mechanisms of and management perspectives on CMT disease.

## 2. Evolving Classification of CMT

Historically, the definition of CMT was anchored on clinical phenotype and nerve pathology observations. However, later electrophysiological examination led to the classifications of CMT into three broad subgroups: demyelinating CMT1, axonal CMT2, and intermediate CMTs [8]. Measuring the action potential of the forearm ulnar motor nerve is the standard method of demarcating axonal versus demyelinating pathogenesis (using the ranges very slow < 15 m/s; slow 15–35 m/s; intermediate 35–45 m/s; and normal > 45 m/s), which is helpful in the identification of inherited neuropathies [9].

CMT is mainly classified into demyelinating CMT1 and axonal CMT2 based on neural conduction velocity and primary defects. An intermediate type-I CMT with a median motor nerve conduction velocity (MNCV) of between 35 m/s and 45 m/s was introduced to describe cases that do not align with the pathological features of CMT1 or CMT2 [10,11]. CMT3 (also called Dejerine–Sottas disease) defines a spectrum of clinically severe diseases that lead to the loss of ambulatory milestones and generalized neurologic deficits, usually beginning in infancy or childhood. A dominantly inherited axonal form of CMT2 with variable pyramidal features (including extensor plantar responses, mild increase in tone, flexor plantar weakness, and preserved or increased reflexes in the knee and ankle) has been reclassified as CMT5 [12,13,14].

The inheritance patterns of CMT have been designated as autosomal dominant (AD), autosomal recessive (AR), and X-linked patterns. CMT1A and CMT2A are AD, while CMT4C is AR. AR demyelinating morphologies result in CMT1 to CMT4, and axonal CMT2 can be AD-CMT2 or AR-CMT2, while I-CMT is divided into dominant (DI-CMT) and recessive (RI-CMT) forms [15]. An intermediate MNCV is often observed in X-linked CMT type 1 cases [16].

Initially, cases of HMSN4 not corresponding to CMT4 were described as the recessively inherited phytanic acid metabolic disorder Refsum disease (involving mutations of phytanoyl-CoA hydroxylase and PHYH ablation of alpha-oxidation and resulting in phytanic acid accumulation), which has childhood onset and systemic multiorgan involvement including ichthyosis, retinitis pigmentosa, and ataxia [17]. Currently, the terms CMT5, CMT6, and CMT7 are generally not used, as clinicians favor the genetic-cause-based designations [4].

CMTX is an X-linked inherited CMT and the second most common form. CMTX involves mutations of *Gjb1* located on the X chromosome; this gene encodes the gap junction protein connexin 32, which is expressed in Schwann cells and oligodendrocytes and forms gap junctions in noncompact myelin, playing an important role in the homeostasis of myelinated axons [18]. The *Gjb1* mutation was first identified in 1993 and accounts for 90% of CMTX1 cases [19]. CMTX is characterized by clinical features similar to classical CMT, with transient central nervous system symptoms such as weakness and dysarthria and reversible white matter lesions [20,21]. CMTX can be classified into six types: CMTX1 and CMTX6 have X-linked dominant patterns, and the remaining four have X-linked recessive patterns [20].

CMT1A is the most prevalent form of Charcot–Marie–Tooth disease, accounting for approximately 50% of all diagnosed cases. It has also been extensively studied and researched [22]. CMTX ranks as the second most common type, representing around 7% to 15% of all cases [23]. CMT2A is the predominant axonal subtype of Charcot–Marie–Tooth disease, accounting for approximately 4~7% of all cases [24]. The remaining types of Charcot–Marie–Tooth disease, including CMT3, CMT4, CMT5, CMT6, and CMT7, occur rarely, with no accurate data available on their prevalence currently.

To date, several causative genes of CMT have been identified, and these are shown in Table 1. Depending on the diversity of genetic mutations that cause the disease, CMT can be further divided into various subtypes; e.g., CMT1A, B, C, or D are associated with *Pmp22* [25], myelin protein zero (*Mpz*) [26], lipopolysaccharide-induced tumor-necrosis-factor-alpha factor (*Litaf*) [27], and early growth response 2 (*Egr2*) [28], respectively. Understanding the genetic factors facilitates our understanding of CMT pathology and the discovery of new treatment strategies. Further possible pathogenic mechanisms and molecular targets of CMT are shown in Figure 2 and Appendix A.

CMT1A is attributed to mutations in the *PMP22* gene. The overexpression of PMP22 forms protein aggregates and correlates with reduced proteasome activity and the accumulation of detergent-insoluble ubiquitinated substrates, which induce apoptosis in Schwann cells [29]. CMT1B is the second most common type of CMT1, resulting from mutations in *MPZ* gene, leading to ER retention, activation of the unfolded protein response (UPR), and disruption of myelin compaction [30]. CMT1E is attributed to mutations in the *PMP22* gene. The leucine-to-proline (L16P) amino acid identical substitution in PMP22 results in Schwann cell hyperproliferation, abnormally thin myelin, axonal degeneration, and subaxonal morphological changes [31]. *MFN2* is the causative gene of CMT2A [32]. Missense mutations of *MFN2* result in the dysfunction of transmembrane GTPase proteins in the outer mitochondrial membrane, followed by mtDNA instability [33]. Altered fusion is associated with an imbalance in mtDNA maintenance and distribution, ultimately impairing OXPHOS [34]. CMT2D is caused by dominant mutations in *GARS*, which alter the conformation of glycyl-transfer RNA (tRNA) synthetase (GlyRS). This alteration enables GlyRS^CMT2D^ to bind the neuropilin 1 (Nrp1) receptor and disturb the VEGF–Nrp1 signaling axis, resulting in selective degeneration of peripheral axons and leading to deficits in distal motor function [35]. CMT2E is caused by mutations in the *NEFL* gene, resulting in reduced neurofilaments (NFs) expression in cutaneous nerve fibers, leading to subsequent alterations in small axonal caliber and reduced velocities [36,37]. CMT2F is caused by mutations in heat-shock protein 27 (Hsp27) [38]. Mutant Hsp27 decreases mitochondrial ceramide levels, resulting in structural and functional changes in mitochondria that lead to hyperphosphorylation of NFs and reduced anterograde transport of NFs [39]. CMT2S is attributed to biallelic mutations of the IGHMBP2 gene, which encodes an ATP-dependent 5′ to 3′ RNA helicase. These mutations disrupt the biochemical association with ribosomal proteins, pre-rRNA processing factors, and tRNA-related species, leading to disturbances in the transcriptional and translational processes [40]. CMT4C is follows an autosomal recessive inheritance pattern resulting from biallelic variants in the *SH3TC2* gene. This gene encodes a protein involved in regulating endosome recycling in myelinating Schwann cells, thereby impacting receptor dynamics [41]. CMT4J arises from mutations in the *FIG4* gene, leading to impaired endolysosomal trafficking in both motor neurons and Schwann cells. This results in severe demyelination and axonal loss, affecting both motor and sensory functions [42]. The pathogenesis of CMTX1 is typically attributed to mutations in the GJB1 gene, which encodes for connexin 32 (Cx32), a gap junction beta 1 protein. This gap junction protein is widely distributed in the peripheral and central nervous systems as well as in the liver, kidneys, and pancreas [19].
biomolecules-14-01138-t001_Table 1Table 1Classification of Charcot–Marie–Tooth.ElectrophysiologyPathophysiologyInheritanceCausative GenesCMT1DemyelinatingNCV < 35 m/sADCMT1*PMP22*, *MPZ*, *LITAF*, *EGR2*, *NEFL*CMT2*KIF1B*, *MFN2*, *RAB7*, *TRPV4*, *P0*, *GDAP1*, *HSPB8*, *DNM2*, *LRSAM1*, *MT-ATP6*, *MARS*, *HARS*CMT2AxonalNCV > 45 m/sARCMT2*LMNA*, *MED 25*, *GDAP1*, *MFN2*, *NEFL*, *TRIM2*CMT4*GDAP1*, *MTMR2*, *MTMR13*, *EGR2*, *SBF1*, *NDRG1*, *FIG4*, *SURF1*, *PRX*, *HK1*I-CMT *Mixed [43]35 < NCV < 45 m/sX-linkedCMTX*GJB1*, *PRPS1*, *PDK3*, *AIFM1*I-CMT: Intermediate CMT; AD = autosomal dominant; AR = autosomal recessive; *PMP22*: Peripheral myelin protein-22; *MPZ*: Myelin protein zero; *LITAF*: Lipopolysaccharide (LPS)-induced tumor necrosis factor (TNF) α factor; *EGR2*: Early growth response 2; *NEFL*: Neurofilament; *KIF1B*: Kinesin family member 1B; *MFN2*: Mitofusin 2; *TRPV4*: Transient receptor potential ion channel subfamily V 4; *GDAP1*: Ganglioside Induced Differentiation Associated Protein 1; *HSPB8*: Heat shock proteins B8; *DNM2*: Dynamin 2; *LRSAM1*: Leucine-rich repeat and sterile alpha motif-containing protein 1; *MT-ATP6*: Mitochondrially encoded ATP synthase 6; *MARS*: Methionyl-TRNA Synthetase; *HARS*: Histidyl-TRNA Synthetase; *LMNA*: Lamin A/C; *MED 25*: MEDIATOR25; *TRIM2*: Tripartite motif-containing protein 2; *MTMR*: Myotubularin-related protein; *SBF1*: SET binding factor 1; *NDRG1*: N-myc downstream-regulated gene 1; *PRX*: Periaxin; *HK1*: Hexokinase 1; *GJB1*: Gap Junction Protein Beta 1; *PRPS1*: Phosphoribosyl pyrophosphate synthetase 1; *PDK3*: Pyruvate Dehydrogenase Kinase 3; *AIFM1*: Apoptosis inducing factor mitochondria associated 1.* Most CMT patients and families with intermediate NCV carry mutations in the *GJB1* gene and have CMTX.


## 3. Available Management Options for CMT Phenotypes

### 3.1. Rehabilitation, Orthoses, and Surgery

Exercise has been identified as an effective treatment strategy for people in the early stages of CMT. To prevent disease progression, rehabilitation interventions including muscle strengthening, stretching, aerobic conditioning, and proprioceptive exercises are recommended [44,45]. Strengthening exercises focus on compensating for the weaker distal muscles, while stretching is employed for joint motion maintenance and contracture prevention [46]. Comprehensive aerobic exercise programs maximize muscle and cardiorespiratory function and prevent additional muscle atrophy in people with neuromuscular disease [47]. 

Proprioceptive exercises significantly improve balance control, gait, functional mobility, and musculoskeletal endurance [48]. Usually, long-term exercise combination programs effectively improve functional activities [45]. Maggi et al. [49] reported that TreSPE (treadmill, stretching, and proprioceptive exercises) physiotherapy, including a 90 min exercise session of 30 min on the treadmill (from 40% to 70% of cardiopulmonary effort test), 10 min rest, 25 min respiratory rehabilitation with positive expiratory pressure-bottle and expiration with glottis open in the lateral position, and 25 min proprioceptive exercises twice a week for 8 weeks, improved lower extremity functional performance. Typical modes of exercise prescription are shown in Appendix A.

Orthoses is another CMT management method that complements rehabilitative approaches and plays an important role in improving balance, gait, and the ability to perform the activities of daily life [44].

As the disease progresses, orthotic foot wear may be required to slow down the development of deformities, supporting against foot drop, stabilizing gait, and reducing pain or sores [50]. An appropriate orthotic device is often important in progressive disorders [51]. Various orthotics devices, ranging from shoe inserts to ankle foot orthoses (AFOs), according to the severity of the disorder as measured by the CMT Neuropathy Score and Ambulation Index, are prescribed to most CMT patients. 

Shoe inserts are usually prescribed first for CMT patients. They help to release pressure on the weight-bearing region to reduce deformities and pain. AFOs should be prescribed when calf muscle weaknesses and foot drop exist, as they help to stabilize the knee during weight-bearing, compensate for a lack of propulsion during the stance phase, and maintain posture and balance [52,53]. According to the experience of physical therapists and the subjective attitudes of patients, custom-fitted orthoses are more comfortable, enable better compliance, and provide better pain relief. 

Over time, the deformities of CMT patients tend to progress in severity [54]. Based on an idea proposed by Glenn B. Pfeffer et al. in 2020 [55], the consensus is that early surgical intervention can minimize progression of cavovarus deformities and soft-tissue contractures.

In the early stages of the condition, flexible deformities including equines, cavus, or forefoot valgus and hindfoot varus can be corrected by joint-sparing osteotomies, soft-tissue release, and tendon transfers, with elaborate preoperative evaluation and individualized plans [56,57,58]. Operative interventions at this stage may prevent the progression of deformities and joint fusions in the future. However, for patients in later stages, who have severely rigid deformities, a complex plan combining ligament and soft-tissue release, osteotomies, and arthrodesis may be an inevitable choice [59]. 

### 3.2. Drug-Based Disease Management

#### 3.2.1. General Symptoms Relief

As physiotherapy, proper foot care, and surgery are not enough to ameliorate CMT-related symptoms, additional symptomatological drugs are used. Pain symptoms in CMT have a dual nature; for example, non-neuropathic pain is linked to skeletal deformities of the foot and spine, leading to altered posture, arthritic degeneration, and neuropathic pain [60]. Generally, NSAIDs (non-steroidal anti-inflammatory drugs), paracetamol, or SSRI (selective serotonin reuptake inhibitor) drugs are prescribed to address the pain [61]. Neuropathic pain is a frequent finding in CMT related to Aδ fiber impairment. Aδ fibers refer to sensory nerve fibers with a myelin sheath, conducting cold, pressure, and pain signals (5~30 m per second) and producing the acute and sharp experience of pain [62]. In CMT diseases, Aδ fibers are tested with a significant abnormal laser-evoked potentials, which may be associated with neuropathic pain symptoms. Calcium channel α2-δ ligands, including gabapentin and pregabalin, which bind to calcium channels, leading to the decreased release of neurotransmitters, have been widely applied in peripheral pain syndromes [63]. Recently, the transient receptor potential vanilloid 1 (TRPV1) channel associated with chronic pain was identified as a prime target in pain management, but adverse effects such as hyperthermia hinder TRPV1 antagonists as novel pain relievers [64].

CMT patients of all subgroups present higher MFI-20 (Multidimensional Fatigue Inventory) scores for fatigue, which might be due to their tendency toward motor weakness, foot deformities, and loss of proprioceptive control [65]. The MFI-20 is a 20-item self-report measurement of fatigue covering the five dimensions of general fatigue (GF), physical fatigue (PF), mental fatigue (MF), reduced motivation (RM), and reduced activity (RA) [66]. Interval-training exercise cycling with creatine supplementation was shown to significantly increase MHC type IIa compositions, improving physiological, neuromuscular, and functional capacities and alleviating fatigue in patients with CMT [67,68]. MHC type IIa is one of the MHC isoforms detected by immunohistochemistry with MHC-specific monoclonal antibodies. As a kind of fast-twitch fiber, MHC type IIa is first detected in neonatal muscles and regulated by neural, hormonal, and mechanical factors in developing [69]. Although no significant correlations between sleep disorders and fatigue have been found in patients with CMT, the analeptic drug modafinil has shown some benefits for treating fatigue but with substantial side effects [70,71].

Drug treatment for CMT primarily focuses on symptom relief, whereas gene therapy aims to ameliorate or potentially cure symptoms of CMT with minimal adverse events by transferring genetic material into the cells of the patients.

#### 3.2.2. Emerging Drug Treatments

Many promising compounds targeting pathophysiological pathways are being tested as CMT treatments. Given the role of PMP22 dosing in the pathogenesis of CMT1A, treatment efforts have focused on reducing PMP22 expression [3]. *Pmp22* transcription is a cAMP-induced action, as two binding sites for cAMP response element binding proteins (CREBs) reside in the *Pmp22* promoter [72]. By modulating the activity of signaling proteins acting upstream of the *Pmp22* gene, ascorbic acid, a specific antioxidant, had an inhibitory effect on adenylate cyclase activity and reduced the production of intracellular cAMP, improving motor function in a CMT1A rat model, but its clinical efficacy is not obvious [3,73]. Similarly, onapristone, a progesterone receptor antagonist, acts upstream of membrane progesterone receptors (a subtype of G protein-coupled receptor), suppressing adenylyl cyclase activity and inducing a decrease in cAMP accumulation [74,75,76]. However, onapristone was suspended due to fatal side effects, which were reported in a clinical trial (NCT02600286).

In addition to target upstream pathways, many downstream signaling pathways such as PI3K-AKT/MEK-ERK, two opposing pathways regulating Schwann cell differentiation, have been reported as promising therapeutic sites, for example, PXT-3003, in the pathomechanism of the disease [77]. In CMT, PMP22 overexpression directly interferes with the PI3K-AKT-mTOR signaling pathway. Subsequently, PI3K-AKT leads to decreased activity of MEK-ERK [78,79]. PXT-3003 is a combination of baclofen (activated on GABA receptors), naltrexone (an agonist of opioid receptors), and sorbitol (a chaperone-binding muscarinic acetylcholine receptor). Activation of the above-mentioned receptors inhibits adenylyl cyclase conversion of ATP into cAMP, which decreases PKA phosphorylating activity and thus ameliorates the PMP22 expression mediated through CREBs. Moreover, cAMP maintains a balance between other signaling pathways such as PI3K-AKT and MEK-ERK [77,78]. 

Neuregulins (NRGs) are a family of growth factors encoded by four genes (NRG1–4), all of which act through ErbB tyrosine kinase receptors and contain EGF-like domains that are essential for activity [80]. NRG1 is by far the most extensively characterized. Schwann-cell-derived NRG-1 type I (NRG1-I) acts via an ErbB2-receptor–MEK/ERK signaling axis and causes onion bulb formations in CMT1A. Blockage of the ErbB2 receptor in adult CMT1A patients may prevent aberrant NRG1 signaling and constitutes a promising avenue for future therapies [81]. Surprisingly, axonal NRG1-I was found to be beneficial for Schwann cell differentiation and axonal preservation in the peripheral nervous system and enhances PI3K-Akt signaling despite endogenous expression of NRG1-I being low in dorsal root ganglion (DRG) and motor neurons [82]. Thus, axonal NRG1-I can drive diseased Schwann cells toward differentiation and preserve peripheral axons, modulating the pathogenesis of CMT1A disease [79,83].

Axonally derived NRG-1 type III (NRG1-III) is another essential modulator of Schwann cells that acts during development, nerve repair, and remyelination. By binding to the cognate ErbB2/3 receptor, NRG1-III activates the PI3K-Akt signaling pathway and modulates the development of peripheral myelination [80,84,85,86]. Therefore, modulating NRG1-III activity may constitute a general therapeutic strategy for treating CMTs characterized by reduced myelination levels. Alpha-secretase tumor-necrosis-factor-alpha-converting enzyme (TACE) cleaves NRG1-III to limit the amount of functional NRG1-III in cells [87]. Neurophysiology and hypermyelination were found to be ameliorated by inhibiting TACE with axonal NRG1-III signaling in a CMT1B mouse model [88]. NRG1-III may also contribute to the hypermyelination of CMT neuropathy [79]. Niaspan treatment hypermyelination is characterized by decreased NRG1 type III and Akt activation [89]. Schwann cell NRG1-I expression is negatively controlled by axonal NRG1-III, which exhibits severely impaired nerve regeneration [83].

Notably, the overexpression of PMP22 can augment the expression of the ligand-gated ion channel P2X7, leading to the opening of purinergic receptor channels and an influx of extracellular Ca^2+^, triggering several Ca^2+^-induced functional effects such as proliferation and differentiation [90]. In this regard, the development of P2X7 inhibitors is expected to provide a new therapeutic strategy for CMT neuropathy [91]. An antagonist of the P2X7 receptor (A438079) was shown to ameliorate the myelination of organotypic DRG neurons and control the biochemical derangements characterizing CMT1A [92].

Recent evidence indicates that changes in the gut microbiome play a role in the development of numerous neurodegenerative disorders through bacterial-related metabolites or inflammatory signals via the gut–brain axis [93,94]. Interventions targeting the gut microbiome have demonstrated promising results in mitigating disease progression, highlighting their therapeutic potential [95]. Mitochondrial dysfunction is the most common mechanism of many types of CMT. Microbiome-derived metabolites such as nicotinamide, colonic acid, and methyl metabolites may regulate the processes of CMT by modulating mitochondrial activity and thereby influencing myelin sheath integrity and axonal transport [96]. For the central nervous system (CNS) phenotype of CMT-X, the ongoing inflammatory response and abundant reactive oxygen species (ROS) synthesis may be the underlying mechanism. Microbial metabolites such as short-chain fatty acids and polyamines found within the gut can influence the immune response and have an impact on the CNS inflammatory state [97,98]. 

#### 3.2.3. Gene Therapies

Gene therapy is the treatment of a disease through transferring genetic material into cells of the patients [99]. Gene addition and gene replacement are important methods for gene therapies. The study by Kagiava A. et al. utilized cx32 gene (with lentiviral vectors) addition for the treatment of CMT1X. Intrathecal gene delivery resulted in normal expression of virally delivered cx32 in T55I KO mice and improvement of the phenotype [100]. Elena Georgiou et al. reported on gene therapy for CMT4C using an adeno-associated viral 9 vector (AAV9) to deliver the human SH3TC2 gene in the Sh3tc2^−/−^ mouse model. This intervention resulted in reduced ratios of demyelinated fibers, increased myelin thickness, and decreased g-ratios at both time points of intervention [101].

Recently, RNA interference (RNAi) techniques have been extensively investigated as potential therapeutic strategies for CMT. RNAi involves the interaction between RNA and corresponding mRNA to form double-stranded RNA (dsRNA) that effectively inactivates the target gene [102]. The mechanisms of common RNAi technology are shown in Figure 3.

ASOs are short synthetic analogues of natural nucleic acids designed to specifically bind to a target mRNA via Watson–Crick hybridization, inducing the selective degradation of the mRNA by RNase or prohibiting the translation of the selected mRNA into protein [103]. In a CMT1A rat model, the overexpression of *Pmp22* was markedly reduced by specifically designed ASOs, and this reversed demyelination and axonal loss neuropathy and restored the electrophysiological properties of peripheral neurons, providing a potential treatment for CMT1A [104]. 

Triplex-forming oligonucleotides (TFOs) are short, single-stranded oligomers that hybridize to a specific sequence of duplex DNA. Instead of binding to mRNA, TFOs were designed to bind to purine-rich target sequences of the two *Pmp22* gene promoters within the duplex major groove to competitively inhibit the binding of transcription factors, proving useful reagents for the therapy of CMT1A [105,106]. 

When cleaved by dicer, hairpin precursor miRNAs (pre-miRNAs) are transformed into double-stranded mature miRNAs [107]. One miRNA strand, the guide strand, which is an approximately 22-nucleotide noncoding RNA molecule, is used for negatively modulating biological activation at the posttranscriptional level via RISC formation. RISC contains the AGO protein, which can bind to and degrade target mRNAs. In Stavrou et al. study, artificial miR871 was packaged into an AAV9 and delivered into a C61-het CMT1A mouse model. Once miR871 was expressed, it then bound on the 3′UTR of Pmp22 mRNA to trigger target mRNA degradation through the RNA-induced silencing complex (RISC) and reduced PMP22 expression in a translatable gene therapy approach [107]. 

Aminoacyl-tRNA synthetases (aaRS) are essential enzymes in translation that act by moving amino acids to cognate tRNAs during protein synthesis. Dominant monoallelic variants of aaRSs have been implicated in CMT2D [108]. To prevent neuropathy in CMT2D, Morelli et al. developed an allele-specific RNAi strategy to reduce the levels of mutant glycyl-tRNA synthetase (Gars) transcripts. They achieved this through the delivery of mutant *Gars*-targeted artificial miRNA expression cassettes packaged within self-complementary AAV9 [108,109].

Using the same RNAi machinery that processes miRNA, a hairpin-shaped shRNA, consisting of a stem region of paired antisense and sense strands and unpaired nucleotide loops, can be converted into an siRNA in the nucleus. SiRNAs are 21- to 23-nucleotide dsRNAs that lead to RNA-induced transcriptional silencing of target mRNA in a sequence-specific manner by RISC in the cytoplasm. As naked siRNAs rarely have effects in vivo, various siRNA delivery systems, such as nanoparticles (NPs), are in development. Using squalene (SQ) NPs, Boutary et al. linked siRNA for *Pmp22* to SQ NPs, which offered protection and improved the stability of the siRNA, for CMT1A treatment via the normalization of disease gene expression [110]. In addition, shRNAs such as shMstn A have been synthesized and introduced into the nuclei of target cells using bacterial or viral vectors to carry out gene expression inhibition in the treatment of CMT disease [111,112]. Gautier et al. successfully acquired a preventive effect on development of pathological features in a CMT1A rat model through the local delivery of AAV2/9 expressing shRNAs against PMP22 [113]. Unfortunately, viral vectors often cause immunologic reactions, which currently limit their use.

Current gene editing technology is laying the foundation for the formulation of future disease therapies [114]. Discovered in the adaptive immune system of bacteria and archaea, the CRISPR-based gene editing system, consisting of Cas9 endonuclease and guide RNA (gRNA), has great potential in the treatment of genetic diseases, including CMT1A, by directly targeting disease-causing genes [115]. By designing a CRISPR/Cas9 system with a TATA-box of *Pmp22*, gene expression of PMP22 can be downregulated to preserve myelin and axons using intraneural injections [116]. 

More details of promising compounds and gene therapeutic strategies for treating different types of CMT are provided in Appendix A.

## 4. Stem Cell Research

Patient-specific induced pluripotent stem cells (iPSCs) can be differentiated into relevant cell type(s) for the in vitro confirmation of disease mechanisms and targets using models that accurately simulate the pathogenesis of CMT disease [117,118]. Although rare clinical treatment with iPSCs has emerged to date, patient-specific iPSCs-based culture systems including microfluidic chips, organoids, and assembloids have been identified as a particularly powerful platform for disease modeling and preclinical studies [119]. An iPSC-derived self-organizing organoid model is more likely to capture biologically meaningful features of the disease and capture the physiological complexity [120]. For CMT1A, iPSC-derived human organoids have been applied to further evaluate amelioration effect on myelin defect by downregulation of PMP22 [121].

Lu et al. reported that an iPSC line derived from the GARS (G294R) family with fibular atrophy was successfully induced, and the mutated gene loci were repaired at the iPSC level using CRISPR/Cas9 technology for the treatment of CMT2D^123^. The surprising results indicated that a combination of CRISPR/Cas9 and iPSCs is a potential therapeutic method for CMT. 

In addition to iPSCs, other types of stem cells, such as Schwann cells derived from dental pulp stem cells (DPSCs), which are typically obtained from third molar extractions and present minimal ethical concerns due to their origin as medical waste, can also be utilized for studying demyelinating neurodegenerative diseases [122].

DPSCs have been used to uncover dysfunction in the endo-lysosomal–autophagy pathway, which is essential for transporting myelin proteins to the plasma membrane in myelinating Schwann cells and contributes to the development of demyelinating neurodegenerative diseases, while retaining their stem cell properties, including high proliferative potential and self-renewal abilities even after cryopreservation [123].

## 5. Therapeutic Education and Genetic Counseling

Proper therapeutic education and genetic counseling are important for patients with CMT and their families. Therapeutic education includes awareness, information, learning, and psychological and social support aimed at helping the patient understand the disease and treatments, participate in care, take charge of his or her state of health, and encourage, as far as possible, the maintenance of daily activities [61]. For instance, the majority of patients with CMT exhibit cavovarus foot deformity. By providing therapeutic education, patients can be guided to modify their daily habits, minimize stair and hill climbing, engage in regular moderate physical activity tailored to muscle capacity, enhance muscle strength and endurance, alleviate fatigue and pain sensations, correctly utilize AFOs for support, and prevent falls. Additionally, psychological counseling is offered to help patients manage tension and anxiety while receiving companionship and psychological support.

Genetic counseling is crucial for CMT patients to comprehend and adjust to the medical, psychological, and familial implications of genetic factors contributing to the disease [124]. Whole-genome sequencing can identify the type and genetic spectrum of CMT. Genetic counseling empowers patients with a comprehensive understanding of the disease’s pathogenesis and inheritance risk to future generations, offering essential guidance for treatment timing and programs. At the same time, genetic counseling has significant societal implications in terms of eugenics.

## 6. Emerging Perspectives for CMT Disease

### 6.1. What Is the Most Suitable Route of Administration for CMT Treatment?

There are various of administration methods for current pre-clinical and clinical treatment on CMTs (drug and gene therapies). Generally, two fundamentally different routes include localized injection versus widespread delivery [125]. For widespread intravascular administration, most sites including non-tented tissues can be reached, but potentially high doses are required, which may cause systemic toxicity. However, for intra-tissue like intraneural and intramuscular tissue, unintended spread can be decreased, but these invasive methods are capable of inducing local reactions such as bleeding and reginal tissue or cells degeneration. In the clinic, intraneural is considered an invasive method that damages the nerves and requires high doses of anesthesia. In oral administration, transduction efficiency is limited by the absorption of the digestive system and clearness of preexisting NAb and clearance in liver [126]. In CNS, intrathecal delivery such as intracerebroventricular (ICV), intracisternal magna (ICM), and intrathecal injection by utilization of fluid dynamics to distribute compounds is another delivery mode. In additional to the invasive adverse, dorsal root ganglia toxicity has been repeatedly found [127]. 

### 6.2. How to Develop Experimental Animal Models for Preclinical Testing

Once the CMT model is established, therapeutic efficacy can be assessed through observation of pes cavus and claw hand recovery, muscle biopsy, electromyography, and gene sequencing. However, these assessments are challenging to perform in small animals. NHPs exhibit developmental trajectories similar to those of humans in terms of their anatomy, physiology, genetics, and neural functions [128]. They also demonstrate similarities in cognition, emotion, and social behavior. Therefore, NHPs may be potential candidates for further pre-clinical testing in CMT disease. However, it is important to note that creating genetically modified NHPs is illegal in the U.S. and Europe, while pigs are allowed to be genetically modified.

If we aim to construct the CMT model of NHPs, CRISPR/Cas9, CRISPR-Cas nucleases, and CRISPR base editor genome editing technologies can be employed. These tools are delivered in vivo using lipid nanoparticles or AAVs to efficiently and precisely modify disease-related genes [129]. While the common administration approach for animal models involves vessel injection, the characteristics of the blood–nerve barrier, which can impede gene transfer to the nervous system, differ significantly from those in humans. This limitation may have hindered our study from achieving optimal conditions for gene delivery. Therefore, exploring various administration methods and doses across different animal models, including mice, pigs, and NHPs, can provide more specific treatment information regarding efficacy and safety. 

CMT is a progressive condition with a long disease duration, making it challenging to determine the appropriate monitoring period post treatment. In our view, lifelong monitoring of treatment effects for CMT should be incorporated into experimental design, and a larger number of experimental animals would be beneficial. This approach is essential due to the numerous types of CMT; increasing the sample size can help mitigate bias and yield more objective results. However, this task is hindered by time and labor costs.

### 6.3. Why Is AAV9 Considered the Most Promising Delivery Vehicle for Gene Therapy?

AAV has been extensively utilized to deliver therapeutic genes into target cells involved in pathological processes of genetic neurological disorders [130]. AAV9 exhibits enhanced capability to target the brain and spinal cord while demonstrating improved blood–brain barrier penetration following minimally invasive intravenous interventions [131]. Meanwhile, AAV9 has the ability to traverse synapses after infection, allowing it to cross synapses and propagate to interconnected neurons. For neuromuscular diseases, it is common to observe conditions that affect both neurons and muscle cells [132]. In some cases of muscular dystrophy, patients have been successfully treated or improved through direct regional injection using genetic strategies [133]. Additionally, novel recombinant adeno-associated viruses (rAAVs) with modifications such as the insertion of a 10-mer peptide fragment from AAV2-Retro into the capsid of AAV9, can retrogradely infect projection neurons and even transport across a wide scale of the central nervous system for clinical treatment of neurological and neurodegenerative disorders such as CMTs [134].

### 6.4. What Factors Contributed to the Failure or Lack of Progress of the Drug Treatments in Clinical Trials?

Among all the aforementioned drug treatments, it is difficult to determine which ones are most likely for future clinical practice. As of now, no definitive pharmacological treatment has been established for any variant of CMT. The concept of pleiotropic therapy aims to achieve a synergistic effect using very low doses of multiple drugs. It is important to consider that any drug administered in this manner would need to be taken for the duration of one’s life, potentially leading to issues related to side effects, compliance, and costs. It is interesting to note that PXT3003, a combination of baclofen (a GABA agonist), sorbitol (a disaccharide), and naltrexone (an opioid antagonist) predicted by computer-based analyses, acts synergistically in down-regulating PMP22 in the most advanced phase [15]. In addition to specific and unknown side effects, the understanding of distinct mechanisms of pathogenesis has led to the identification of small molecules that may have therapeutic benefit [135].

### 6.5. How to Evaluate the Risk of Novel Treatments

Based on preclinical data, observations can be made regarding potential side effects of these novel treatments. Additionally, considerations such as immunogenicity and genotoxicity arising from novel transgene approaches are essential issues that need to be taken into account. Improving potency and selecting the optimal approach to achieve therapeutic efficacy are critical issues to consider before deciding to participate in a clinical trial. Many studies have shown that large primate animal models exhibit approximately 50–100 times less transduction compared with mice, which are the most commonly used disease animal model [136]. While CMTs are not fatal and are lifelong diseases, it is crucial to gather essential knowledge on the safety of novel treatments through preclinical observations in order to achieve the goal of long-term correction for these genetic and metabolic diseases. Therefore, it is critical to identify therapeutic biomarkers that can indicate the progression of this slow disease before conducting a clinical trial. Additionally, the significant cost of novel treatments should be taken into consideration before preparing for further transformation trials.

## 7. Conclusions

Scientific knowledge of CMT mechanisms continues to evolve, and it is now clear that the dysregulation of myelin and axons is key in the pathogenesis of CMT. In the past two decades, considerable progress has been made in our understanding of the importance of NGS for identifying targets within pathways that lead to CMT.

Single-target treatments do not typically achieve satisfactory therapeutic effects for complex diseases involving multiple factors [137]. CMT has multiple pathogenetic mechanisms, whose interplay and feedback loops may require not one but multiple combinations treatments. However, achieving treatment innovations in the repertoire of the pathways involved is a formidable mission, underscoring the urgency for transformative strategies [138].

The currently used management methods of CMT symptoms are often associated with poor efficacy and side effects in normal cells and organs [139]. In the development of novel strategies of neuron-, Schwann cell-, and axon-targeting drug delivery, the application of a variety of biodegradable, bio-responsive, and targeted nanocarriers, including polymeric nanoparticles, liposomes, extracellular vesicles, inorganic NPs, cell-targeting viral capsids, and hydrogels, have been promising approaches that provide efficient site-specific delivery and reduced systematic side effects [140]. Additionally, genetic therapy strategies using agents such as siRNA, shRNA, miRNA, and ASO have been widely used to treat CMT [141]. Delivery vehicles for these therapeutic oligonucleotides may also improve agent stability and transformation efficiency when targeting pathogenicity [142,143].

In conclusion, the management of CMT involves a comprehensive approach based on an understanding of pathogenesis and disease progression, encompassing pharmacotherapy, gene therapy, patient education, rehabilitation, and orthopedic surgery. Although numerous gene therapy agents for CMT1A are currently undergoing clinical trials, significant challenges remain. Nevertheless, there is optimism that effective treatments for various forms of CMT will soon become available.

## Figures and Tables

**Figure 1 biomolecules-14-01138-f001:**
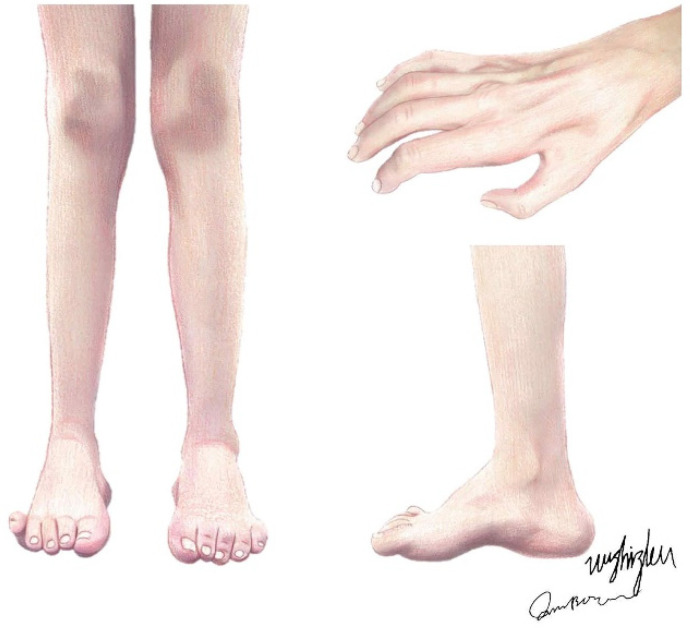
The typical clinical presentations of CMT patients. Claw hands and interosseous muscle atrophy; muscle atrophy of the anterior tibial, peroneal, and posterior tibial muscles; and bilateral pes cavus, i.e., hammertoes.

**Figure 2 biomolecules-14-01138-f002:**
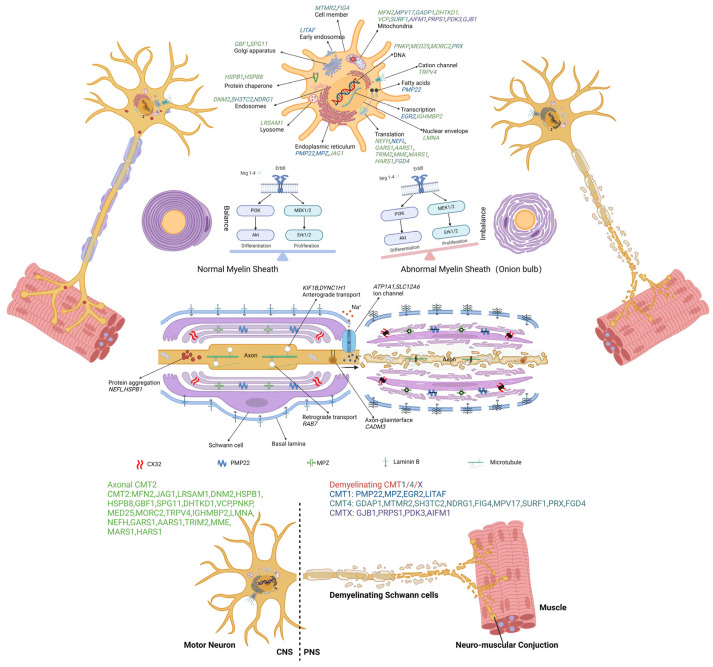
The molecular and genetic targets involving pathogenesis in CMT. The causal genes disturb functions of cellular processes, which are implicated in the pathogenesis of various types of CMT: cell member (*MTMR2*, *FIG4*), early endosomes (*LITAF*), endosomes (*DNM2*, *SH3TC2*, *NDRG1*), lysosome (*LRSAM1*), endoplasmic reticulum (*PMP22*, *MPZ*, *JAG1*), Golgi apparatus (*GBF1*, *SPG11*), mitochondria (*MFN2*, *MPV17*, *GADP1*, *DHTKD1*, *VCP*, *SURF1*, *AIFM1*, *PRPS1*, *PDKS*), DNA (*PNKP*, *MED25*, *MORC2*, *PRX*), transcription (*EGR2*, *IGHMBP2*), translation (*NEFH*, *NEFL*, *GARS1*, *AARS1*, *TRIM2*, *MME*, *MARS1*, *HARS1*, *FGD4*), nuclear envelope (*LMNA*), fatty acids (*PMP2*), and protein chaperone (*HSPB1*, *HSPB8*). The dysregulation of Schwann cell differentiation and proliferation leads to demyelination. Protein aggregation (*NEFL*, *HSPB1*), axonal transport deficits (*KIF1B*, *DYNC1H1*, *RAB7*, *CADM3*), and ion-channel (*ATP1A1*, *SLC12A6*) dysfunction have also been implicated in different types of CMT.

**Figure 3 biomolecules-14-01138-f003:**
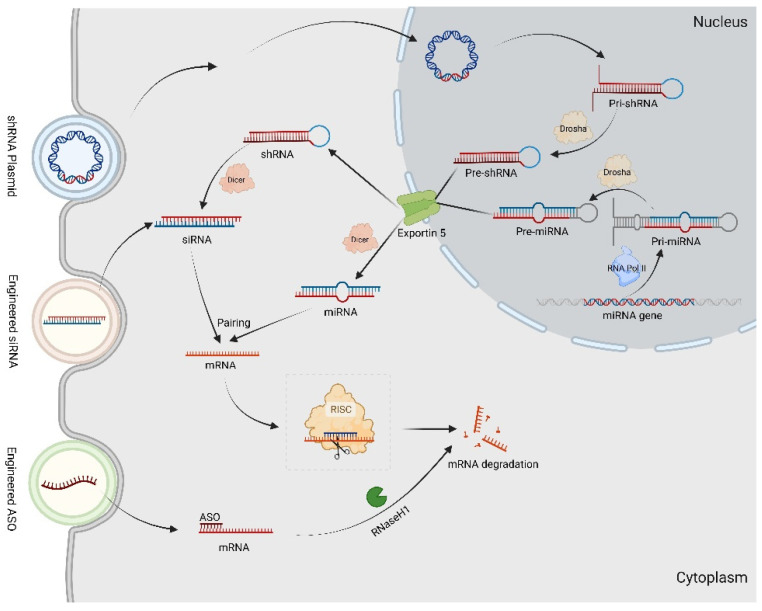
The RNAi technology in gene therapy for CMT. The Pri-miRNAs are transcribed by DNA-dependent RNA polymerase II (Pol II) transcriptional complex. After expression in the nucleus, Pri-shRNAs and Pri-miRNAs are processed by Drosha to Pre-shRNAs and Pre-miRNAs. Afterwards, both Pre-shRNAs and Pre-miRNAs are exported by Exportin-5 to the cytoplasm. The shRNAs and Pre-miRNAs are associated with Dicer, resulting in the removal of the loop sequence and synthesis siRNAs as well as miRNA. By targeting complementary sequence mRNAs, they construct into RISC, resulting in mRNAs degradation. The synthetic ASO combined with target mRNAs generates a DNA–RNA heteroduplex in cytoplasm, which is recognized by RNase H1 to cleave.

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
