# Peer review of "Current Treatment Methods for Charcot–Marie–Tooth Diseases"

_biomolecules, 2024, doi:10.3390/biom14091138_

Round 1

Reviewer 1 Report

Comments and Suggestions for Authors

In the current paper Dong et al., try to give an overview of the state of the art of CMT neuropathies. The main issue is the lack of originality of the paper. There are several papers similar to these review presenting in a more cohesive way similar data. The title is not clear and is not informative-descriptive for what is included in the paper. Further, the authors although in the title and in the first two parts 1 and 2 refer to different CMT types when it comes to the treatment they focuse on CMT1A and the rest are just mentioned in the table S6.

Author Response

Thank you for your careful and patient review of our work.We have revised the title and added sections from various aspects.Please see the attachment.

Reviewer 2 Report

Comments and Suggestions for Authors

Thanks to authors for investing time putting this review together. Their figures and supplementary tables are very well designed and comprehensive.

To my humble opinion, the manuscript is generally well written (minor re-organization suggestions), however it needs a feature that will make it standout among already existing literature. The paper “Emerging Therapies for Charcot-Marie-Tooth Inherited Neuropathies” at IJMS (MDPI) was published at 2021 stating every available drug and gene therapy option for CMTs, at the time. After that many smaller reviews were published on CMTs and their treatment [PMIDs: 38452947, 38712465, 38873808, 38678519, 36965137, 37455204, 36926710, 37808507, 36855793, 36588067]. You need to find something to make your paper of an added value. I also suggest running to these references for cross-checking your statements as there were cases when original papers were not interpreted correctly.

Also, from the title and abstract I was expecting that we will be touching all CMTs. However, you gave background info for a greater spectrum of CMTs, and stated emerging treatments only for CMT1A. If this manuscript was intended to be exclusively focused on CMT1A then the title, abstract and intro should re-adjust. Nevertheless, there are still important CMT1A treatments options that are not included in your main manuscript. You can use above reviews, to see what other reviews usually highlight as important treatments. If according to your opinion, the treatments you included are the most promising, it will be nice including a section with a critical evaluation explaining what makes the rest of the treatments lees appealing.

If you will keep CMT1A as the main focus of your paper, I suggest moving the CMT1A section of TableS6 in the main text.

Overall, I suggest improving the title, based on the final manuscript content. To me “Administration of Charcot–Marie–Tooth Disease: State of the Art” is not a good option. First, nothing new or state of the art was included in this review. Also, having administration in the title I would expect something like “Administration methods to treat Charcot–Marie–Tooth Diseases”.

Suggestions for chapters to include in your manuscript to make it more of an “added value”. You don’t need to follow these suggestions but you definitely need to add something that will make your manuscript stand out.

1.       Comment on current pre-clinical and clinical administration methods to treat CMTs (drug and gene therapies): intrathecal, intravenous, intraneural, intramuscular, oral etc. What are their pros and cons and how effective they are for target engagement (i.e.  getting in Schwann cells, neurons, axons) (Useful material: DOI: 10.18609/nai.2024.011, 38256124)

2.       How big animals (NHPs, pigs etc) can be used to further pre-clinical testing? What conclusions can be drawn when testing potential therapeutic products in wild type animals that do not have the disease and what we should keep in mind when extrapolating the results? Considerations: 1) body size 2) NHPs do not expresses the CMT-causing mutation 3) blood-brain and blood-nerve barriers are intact in normal animals, how about CMT patients? 4) how relevant is testing article-related immunotoxicity in normal animals that do not have disease-related inflammation? 5) how long NHPs/big animals are monitored after treatment? How much should be enough? 6) CMT patients have big roots and onion bulbs in their peripheral nerves, how relevant is testing biodistribution in big animals missing these features?

3.       Gene therapies are an emerging treatment option for CMTs. Refer to existing knowledge from GAN and SMA gene therapies and comment on how these encouraged CMT-experts to consider gene therapies as a treatment option. Also, it seems like AAV9 is the most promising delivery vehicle, which clinical data make AAV9 promising? (Useful material: PMID: 38565561, 38836969, 37766208; ASOs: PMID: 37766208)

4.       For drug treatments, all you drug treatments you cited either failed or never progressed in clinical trials. You can comment on that.

5.       Things to consider before deciding to participate in a clinical trial? Most CMTs are not fatal how you comment on the risk of novel treatments?

Abstract is not clear if the review will summarize, drug treatments, gene therapy treatments or both.

I am not sure if it’s a good idea including so many details on classification protocols/practices that are not used any more.

L15: Not all CMT genetic mechanisms were discovered with NGS, this is more true for the rare types that have been recently discovered.

L17: Please rephrase to: “Currently, there no approved treatments….”

Note: not all proposed pre-clinical treatments are drug treatments, there are also many gene therapy treatments (which are not classified as drug treatments)

L18: Please replace “treatments” with “disease management methods”

L19: please include gene addition i.e. “gene silencing, gene addition and gene editing”.

L20: The only gene therapy that reached clinical trials is for CMT2S. All the rest of the treatment that reached clinical trials were drug treatments. Please correct this sentence accordingly.

L34-39: Please indicate that these are the usual phenotypes of the disease as there are also some CMT types with much more severe phenotypes.

L44-45: Feel like being duplicated in section 2. Merge it with “Evolving classification of CMT” chapter

L46: Please include intermediate CMTs too.

L55: Pmp22 duplication is usually detected with MLPA not NGS

L59: what do you mean by oral drug? Pain killers? Please specify

L61-62: There are not solid evidence supporting that long-term use of ASOs in humans is safe. Please remove this statement.

L61-67: This is a specific treatment approach for CMT1A, it better in the main text not the introduction.

L83-85: Merge with the previous paragraph

L86: Inheritance should be a separate paragraph.

L87: This statement is wrong. For example: CMT1A and CMT2A are AD; CMT4C is AR. Your table 1 is a correct presentation of inheritance. If needed visit “Stavrou et al, 2023, Charcot-Marie-Tooth neuropathies: Current gene therapy advances and the route toward translation”

L105: Please state the prevalence (% of cases) also for the rest of CMT types mentioned above

L118: 2 commas (,,)

Figure 2: Very nice figure! Is it possible to also incorporate CMT types associated with the CMT-related proteins? Also, needs a more detailed legend explaining what see in the figure.

L 138: Please replace treatment with “Available management options for CMT phenotypes”

L150: extra (r)

L157: Please replace the word treatment with management method. You may find this link useful for more info on orthos use https://heliosbracing.com/cmt-guide/

L161: Orthos are not preventive, they provide support while using them.

L184: Replace the word treatment with disease management

L185: Symptoms

L186: Pain killers do not treat causative factors either, only gene therapies can do that. Please replace “However, physiotherapy, proper foot care, and surgery are not enough to ameliorate CMT-related symptoms additional symptomatological drugs are used”

L189: Is not clear how “but” is used in this sentence

L190: Abbreviate NSAIDs and SSRI

L191: Briefly explain Aδ fiber impairment

L199: Briefly explain MFI-20 scores

L202: Briefly explain MHC type IIa compositions

L206: Before describing drug and gene therapy treatments for CMT you need an introductory paragraph saying the difference between the two.

L207: Please change the title to “Emerging drug treatments”

L209-215: Explaining the pathogenesis of each CMT type should be done before describing available treatments. Add a section that will briefly explain the pathogenesis of each CMT type included in treatments section

L219: Onapristone clinical trials have been suspended due to fatal side effects. Please include this info. Also side the clinical trial number (NCT02600286))

L224: Are you referring to PXT-3003?

L266: Gene therapy also includes gene addition; CMT1X CMT4C are only a couple of examples when adding a gene rescue the CMT phenotype pre-clinically.

L297-299: This sentence is wrong. Stavrou et al packaged miR871 into AAV9 vector. This was delivered into mice and once miR871 was expressed it then bound on PMP22 (human) and Pmp22 (murine) mRNAs. The AAV9 vector DID NOT host the Pmp22 mRNA bound to miR871.

L315: Is pharmacological efficacy the wright words to use here? How about using stability instead?

L316: The most important paper on treating CMT1A with shRNAs is Gautier et al 2021 (PMID: 33883545). I suggest you include it in the manuscript.

L330: Important papers to include: PMID: 34128983, 38825644, 36511878, 38672423

L336: There is no red.123 in you list, the last reference the main text is103.

L338: This is a chapter that no other published review has touched. You can extrapolate more.

L350&353: Replace drugs with treatments

L356: Gut microbiome appears for the first time in the conclusions. I suggest moving it to “emerging drug treatments” section and re-adjust the paragraph to show how this can be a treatment option

L364: Please clarify what you mean by Conventional drug therapies. Are you referring to the currently used management methods of CMT symptoms?

L366: Actually CMT1A, need Schwann cell targeting. Better rephrase to neuron, SC and axon targeting.

L369: There is also an emerging field of cell-targeting viral capsids. You can include that too.

L373-379: I am not sure if this is good as a last paragraph. I suggest replacing it with something more relevant to the topic covering most of your manuscript which is CMT1A treatments

Author Response

Thanks for your constructive advice. We have carefully revised the manuscript. Please see the attachment.

Reviewer 3 Report

Comments and Suggestions for Authors

Minor error: 

Line 121: do you mean 'table S1-S4' as supplemental tables 2, 3, 4 are not provided with this article? Kindly check.

It would help in this article if there is a table of all old and new treatments, their target action and level of research done for example basic science/ animal/ human evidence. Failed trials as that of ascorbic acid is also worth including in this table.

In conclusions, you mention about gut microbiome which has not been discussed in the main text. Kindly discuss in main text or remove that paragraph as it is not a concluding statement.

Would you be able to shed light as to how the current classification based on NCV is helping to choose the right treatments. 

Author Response

Thank you for your very valuable advice. We have responsed and revised our manuscript according to your advice.Please see the attachment.

Round 2

Reviewer 1 Report

Comments and Suggestions for Authors

Although the authors tried to improve the text there are still issues that do not allow to accept the paper for publication.

First of all the title is not appropriate and it is misleading regarding the context of the paper. The paper is not discussing only the routes of administration but different aspects of the CMTs including the genetics, symptoms, classifications, drug and gene therapies etc. So the title is not informative for what is included in the paper as I already mentioned in my previous comments. Further, the title is not a proper title.

Further, although the authors added several parts there is lack of cohesion between these parts and the previous text.

Considering the discussion is not a discussion since they are still refering to different treatments and ways of administration mostly for gene therapies.

Author Response

We greatly appreciate your comments on our manuscript. Please see the attachment.

Reviewer 2 Report

Comments and Suggestions for Authors

Thank you for trying to revise the manuscript. Still think that the title is not appropriate. I suggest something like: "Summary on CMT emerging treatments", or something similar. The word administration is misleading as the reader is expecting details on delivery routes.

Move L222-256 after L 118 i.e. before Fig 2.

L328: Start with a sentence defining gene therapies.  

L329: What vehicle Kagiava used for adding Cx32? AAV9?

L368: Please specify “artificial miR871”

L369: Please clarify that “it bound on the 3’ UTR of human and murine PMP22/Pmp22 mRNA.

L451: Are you sure that infection is the right word? I suggest using injection

L454: In the clinic intraneural is considered an invasive method that damages the nerve and requires high doses of anesthesia. You can include this point too.

L456: Replace routine with administration

L460: remove essential

L462: This section needs to be improved with more specific suggestions on experimental design

 L467: In the US and Europe, NHP disease models are not allowed i.e. creating a genetically modified NHP is illegal. Pigs are allowed to be genetically modified.

 Thank you for trying to revise the manuscript. Still think that the title is not appropriate. I suggest something like: "The route towards CMT treatment", or something similar. The word administration is misleading as the reader is expecting details on delivery routes.

Move L222-256 after L 118 i.e. before Fig 2.

L328: Start with a sentence defining gene therapies.  

L329: What vehicle Kagiava used for adding Cx32? AAV9?

L368: Please specify “artificial miR871”

L369: Please clarify that “it bound on the 3’ UTR of human and murine PMP22/Pmp22 mRNA.

L451: Are you sure that infection is the right word? I suggest using injection

L454: In the clinic intraneural is considered an invasive method that damages the nerve and requires high doses of anesthesia. You can include this point too.

L456: Replace routine with administration

L460: remove essential

L462: This section needs to be improved with more specific suggestions on experimental design

 L467: In the US and Europe, NHP disease models are not allowed i.e. creating a genetically modified NHP is illegal. Pigs are allowed to be genetically modified.

Comments on the Quality of English Language

Fine

Author Response

Thanks for your carefully reading and the constructive comments.We have revised the manuscript according to your advice.Please see the attachment.
